# Evolution of Target-Site Resistance to Glyphosate in an *Amaranthus palmeri* Population from Argentina and Its Expression at Different Plant Growth Temperatures

**DOI:** 10.3390/plants8110512

**Published:** 2019-11-16

**Authors:** Shiv Shankhar Kaundun, Lucy Victoria Jackson, Sarah-Jane Hutchings, Jonathan Galloway, Elisabetta Marchegiani, Anushka Howell, Ryan Carlin, Eddie Mcindoe, Daniel Tuesca, Raul Moreno

**Affiliations:** 1Herbicide Bioscience, Syngenta Ltd, Jealott’s Hill International Research Centre, Bracknell RG42 6EY, UK; lvjackson17@hotmail.co.uk (L.V.J.); sarah-jane.hutchings@syngenta.com (S.-J.H.); jonathan.galloway@syngenta.com (J.G.); e.marchegiani@gmail.com (E.M.); anushka.howell@gmail.com (A.H.); eddie.mcindoe@syngenta.com (E.M.); 2Syngenta, Research Triangle Park, NC 27709, USA; ryan.carlin@syngenta.com; 3Cátedra de Malezas, Facultad de Ciencias Agrarias, Universidad Nacional de Rosario, Zavalla S2125ZAA, Argentina; dtuesca@unr.edu.ar; 4Syngenta Argentina, Oficina Central, Av. Libertador 1855, Vicente López, Buenos Aires B1638BGE, Argentina; raul.moreno@syngenta.com

**Keywords:** *Amaranthus palmeri*, Argentina, glyphosate, resistance mechanism, P106S *EPSPS* mutation, *EPSPS* over-expression

## Abstract

The mechanism and expression of resistance to glyphosate at different plant growing temperatures was investigated in an *Amaranthus palmeri* population (VM1) from a soybean field in Vicuña Mackenna, Cordoba, Argentina. Resistance was not due to reduced glyphosate translocation to the meristem or to *EPSPS* duplication, as reported for most US samples. In contrast, a proline 106 to serine target-site mutation acting additively with *EPSPS* over-expression (1.8-fold increase) was respectively a major and minor contributor to glyphosate resistance in VM1. Resistance indices based on LD_50_ values generated using progenies from a cross between 52 PS106 VM1 individuals were estimated at 7.1 for homozygous SS106 and 4.3 for heterozygous PS106 compared with homozygous wild PP106 plants grown at a medium temperature of 24 °C day/18 °C night. A larger proportion of wild and mutant progenies survived a single commonly employed glyphosate rate when maintained at 30 °C day/26 °C night compared with 20 °C day/16 night in a subsequent experiment. Interestingly, the P106S mutation was not identified in any of the 920 plants analysed from 115 US populations, thereby potentially reflecting the difference in *A. palmeri* control practices in Argentina and USA.

## 1. Introduction

Palmer amaranth (*Amaranthus palmeri* S. Wats.) is native to the arid regions of the southwest United States and northern Mexico [1]. It is a tall, erect, annual, summer species capable of attaining heights of two metres [2]. It was once cultivated and eaten by Native Americans including the Navajo, Pima, Yuma and Mohave for its highly nutritious leaves, stems and seeds [3]. From being a marginal and relatively localised species until recently, Palmer amaranth has now invaded vast acreages of cropland in southeast United States to become one of the most troublesome weeds of field corn, cotton, peanut and soybean [4]. Several attributes have contributed to the success of *A. palmeri* as a highly adaptable and invasive weed. It is a C4 species with a very high photosynthetic rate and water use efficiency [5,6]. Palmer amaranth is a prolific seed producer, with single female individuals capable of generating up to 600,000 seeds [7]. *A. palmeri*’s large and deep root systems, its rapid seed germination and seedling growth make it a formidable competitor of warm season crops and other *Amaranthus* species alike [8,9]. When Palmer amaranth co-emerges with soybean, a single plant per 30 cm of row reduced grain yield by as much as 64%, according to a field study in Arkansas [10].

*A. palmeri* is a dioecious weed and thus favours frequent gene exchanges between individuals [11]. The ensuing large genetic variability harboured among *A. palmeri* plants is a major contributor for resistance evolution following herbicide selection pressure [12,13]. Resistance is documented to all major herbicide sites of action that are commonly employed to manage Palmer amaranth. The first reported case of herbicide failure in *A. palmeri* was to microtubule inhibitors in the late 1980s [14]. This was sequentially followed by populations that were no longer satisfactorily controlled with photosystem II (PSII), acetolactate synthase (ALS), 5-enolpyruvylshikimate-3-phosphate synthase (EPSPS), protoporphyrinogen oxidase (PPO) and 4-hydroxyphenylpyruvate dioxygenase (HPPD) inhibiting herbicides [15]. Of concern is the gradual accumulation of resistance to herbicides belonging to different sites of action, further limiting weed control options for *A. palmeri* [16]. Resistance within and between adjacent fields can be transferred by seeds and by the copious amounts of *A. palmeri* pollen that can travel to distances of up to 300 metres and potentially farther [17]. Long-distance spread, on the other hand, is generally achieved by the transfer of seeds in farm machinery, animal feed, manure and, especially, crop seed contaminations [18,19]. In the USA alone, Palmer amaranth is present in at least 28 States, with Northernmost regions being represented by North Dakota and Minnesota [15]. Palmer amaranth is also reported in countries as far as Japan and Australia, Spain, Austria and across Israel and Cyprus, where it is a well-established species [20]. Outside the United States, herbicide-resistant Palmer amaranth populations are only described in Argentina, Mexico, Israel and Brazil [21,22,23,24].

In Argentina, Palmer amaranth was formally identified in the provinces of Cordoba, San Luis and La Pampa in 2013, although it may have been present in the country as early as 2004 but it was confounded with the native *Amaranthus quitensis* species [25]. The *A. palmeri* populations are suspected to have been introduced to soybean fields as a contaminant of seed import from the USA [26]. In a relatively short period of time, it has expanded to a large proportion of soybean production systems, with most populations being resistant to glyphosate [19,27]. *A. palmeri* is considered a real threat to soybean agro-systems in the country, requiring increasingly complicated and costly alternative herbicides and cultural methods for effective weed control [27]. Some of the populations are also resistant to ALS herbicides, thereby significantly limiting post-emergence control options to just PPO herbicides [23].

Given the propensity of *A. palmeri* to evolve resistance to multiple herbicide sites of action, it is imperative to quickly detect new cases of recalcitrant populations and determine the mechanism of resistance involved, with a view to design sustainable management strategies. In spite of the wide distribution and rapid spread of Palmer amaranth in Argentina, there has been no attempt to investigate the mechanism of glyphosate resistance in the populations. The objective of this study was to determine the mechanism of resistance to glyphosate in an *A. palmeri* population (VM1) collected from a soybean field from Cordoba province. Subsequently, we investigated the relative expression of the mechanisms of resistance identified in VM1 using pre-characterised wild and mutant progenies, applied with glyphosate and allowed to grow at different temperatures relevant to soybean cropping systems in Argentina. Finally, we assessed the prevalence of the target-site resistance mutation identified in VM1 in a large number of native US *A. palmeri* populations, which are generally managed at optimal plant growth stages and with a more diverse set of herbicides than in Argentina.

## 2. Results

### 2.1. Resistance Confirmation Test

The standard sensitive *A. palmeri* population ApS was killed at 400 g ai ha^−1^ glyphosate and above under our test conditions (Figure 1). At this rate, 65% of the plants from the known resistant population ApR, characterised by *EPSPS* gene duplication and 80% of plants from VM1 survived the glyphosate treatment. At the commonly used rate of 800 g ai ha^−1^ glyphosate, the survivorship for VM1 and ApR were 45% and 35%, respectively. Only one VM1 and three ApR individuals out of 20 assayed survived at the 2X field rate of glyphosate, whilst all plants were killed at the highest rate of 3200 g ai ha^−1^ glyphosate applied. It is noteworthy that all survivors at 400 g ai ha^−1^ and above were considerably stunted compared with the untreated controls, suggesting a relatively weak resistance mechanism to glyphosate in VM1.

### 2.2. Mechanism of Resistance to Glyphosate

#### 2.2.1. *EPSPS* Gene Sequencing

PCR amplified an expected 195 bp fragment encompassing *EPSPS* codons 54 and 119. The nucleotide sequences showed an average 99% homology with previously published *A. palmeri* data (e.g., GeneBank references: KC169785 and FJ861243), supporting the identity of *EPSPS* amplified here. Three nucleotide substitutions were observed among the 40 plants analysed. These consisted of two synonymous changes at the third base of *EPSPS* codons 82 and 105, which were present in both ApS and VM1 samples. Additionally, a cystosine to thymine transition (CCA to TCA) at the first base of codon 106, resulting into the proline to serine amino acid change was identified in the majority of VM1 plants. The genotypic frequencies at *EPSPS* codon 106 for population VM1 were: 0.05 homozygous wild-type PP106, 0.45 heterozygous mutant PS106 and 0.50 homozygous mutant SS106. As expected, all 20 plants from the standard sensitive population ApS contained the wild-type PP106 allele at the homozygous state.

#### 2.2.2. Level of Resistance Conferred by the P106S *EPSPS* Mutation and Other Potential Glyphosate Resistance Mechanisms in VM1

The progeny (denoted VM1-P) respectively segregated into a 1.00:2.15:0.85 ratio for PP106, PS106 and SS106 individuals among the 1056 plants analysed. Since the VM1-P samples were randomly selected for use in the dose response test, the proportions of PP106, PS106 and SS106 individuals were variable among the different glyphosate rates applied. Nonetheless, the relatively large number (96 plants per herbicide rate) of progeny utilised allowed for an observed minimum of 15 PP106, 40 PS106 and 15 SS106 plants to be tested per glyphosate rate. As with the initial resistance confirmation test, all ApS-sensitive individuals were killed at 400 g ai ha^−1^ and above. At this discriminating glyphosate rate, 26 out of 27 PP106 homozygous wild-type plants from VM1-P were killed (Figure 2). In contrast, 75%, 88% and 93% of PS106, SS106 and ApR plants survived at 400 g ai ha^−1^ glyphosate. Importantly, all of the PP106 plants died, whilst survivorship was recorded at 40%, 73% and 71%, respectively, for PS106, SS106 and ApR individuals, at the commonly employed rate of 800 g ai ha^−1^ glyphosate.

Dose-response analysis estimated LD_50_ values at 67 g ai ha^−1^ for the standard sensitive population ApS to 1102 g ai ha^−1^ for the homozygous mutant SS106 through to 672 g ai ha^−1^ for the heterozygous mutant PS106 plants (Table 1a). The corresponding resistance indices for PS106 and SS106, computed relative to PP106 plants originating from genetically comparable VM1-P individuals were 4.3 and 7.1, respectively (Table 1b).

It is noteworthy that a 2.3-fold resistance increase was identified between the wild-type subpopulation PP106 vs. the standard sensitive population ApS, potentially suggesting minor supplementary underlying resistance mechanisms to glyphosate in VM1. Consequently, the resistance indices for the SS106 and PS106 subpopulations compared with the standard sensitive population ApS were, respectively, 16.4 and 10.0, due to the additive effect of resistance attributed to the P106S mutation and a minor additional glyphosate resistance mechanism in VM1. The resistance indices for the homozygous mutant SS106 subpopulation and the standard resistant ApR characterised by gene duplication, compared with the ApS sensitive standard *A. palmeri* population, were similar at 16.4 and 14.6, respectively.

#### 2.2.3. *EPSPS* Target Gene Duplication and Over-Expression

*EPSPS* target gene duplication and over-expression were evaluated for the standard resistant ApR and the VM1 populations compared with the standard sensitive ApS sample. The individual *EPSPS* gene values are graphically shown on a scatter diagram (Figure 3), and the corresponding averages and statistical analyses summarised in Table 2.

Low levels of variation in *EPSPS* gene copy numbers relative to both *ALS* and *CPS* reference genes were recorded among the 16 ApS plants analysed (Figure 3). The *EPSPS* copy number varied appreciably from 4- to 17-fold relative to the two reference genes for the standard resistant ApR individuals. The estimated *EPSPS* gene copy number ratios between ApR and ApS were highly significant (p < 0.0001) at 9.1 and 8.7 when normalised to the *ALS* and *CPS* genes, respectively (Table 2). Analysis of the VM1 population also indicated a low variability in *EPSPS* gene copy numbers among individual plants. The relative *EPSPS* copy number ratios between VM1 and ApS plants were not significant and evaluated at 1.13 (p = 0.15) and 1.17 (p = 0.16) with respect to the *ALS* and the *CPS* genes. The estimated *EPSPS* gene expression ratios between the known standard ApR and ApS populations were highly significant at 9.9 and 8.4 relative to the *ALS* and *CPS* references, thereby matching the gene duplication results. The relative *EPSPS* expression levels of individual VM1 plants varied from 1.1- to 4.8-fold and 1.1- to 3.9-fold for the *ALS* and *CPS* genes, respectively. The *EPSPS* gene expression ratios between VM1 and ApS were 1.86 and 1.78 when normalised to the *ALS* and *CPS* genes, respectively. Although much smaller in magnitude than the ApR versus ApS comparison, both of the two latter ratios were highly significant (p < 0.0001), suggesting a contribution of *EPSPS* gene over-expression to resistance to glyphosate in VM1.

#### 2.2.4. Glyphosate Uptake and Translocation

The amount of glyphosate uptake varied between 62% and 72% of percentage applied within the time course experiment (Figure 4a). There was no significant difference in the amount of glyphosate absorbed between VM1 and ApS plants (P = 0.28). Likewise, similar amounts (P = 0.41) of radiochemicals were recovered in the meristematic tissues of the ApS and VM1 *A. palmeri* populations (Figure 4b and Table 3). On the other hand, the amounts of glyphosate in the treated leaf and rest of plant sections showed some evidence of time-dependent differences, with the measurements in the two tissues mirroring one another (Table 3). Larger quantities of glyphosate were detected in the treated leaves, particularly at 72H, of the sensitive compared with the resistant population. Analysis of the rest of plant section showed the inverse of this trend, with the differences being similar in magnitude.

### 2.3. Relative Control of Wild and Mutant Plants at Low and High Temperatures

All untreated plants survived at a low temperature of 20 °C/16 °C day/night and a high temperature of 30 °C/26 °C day/night conditions. At assessment time three weeks after treatment, however, the plants kept at the high temperature were on average three times taller than those at the low temperature (data not shown). All standard sensitive *ApS A. palmeri* plants were killed when applied with glyphosate and kept at the low temperature (Table 4a). On average, over the two experiments, 6.5% PP106, 29.9% heterozygous PS106-VM1-P and 42.5% homozygous SS106-VM1-P mutant plants survived but were highly stunted. Analysis of the survival data with the Cochran–Mantel–Haenszel (CMH) test showed a clear demarcation between SS106-VM1-P (P < 0.0001) or heterozygous PS106-VM1-P (P < 0.0001) compared with wild-type homozygous PP106-VM1-P individuals (Table 4b). The differences between SS106-VM1-P and PS106-VM1-P and between PP106-VM1 and PP106-ApS were smaller in magnitude though still statistically significant (P = 0.0291 and P = 0.050, respectively). All ApS *A. palmeri* plants treated with the single rate of glyphosate also died when maintained at the high temperature (Table 4a). Seventeen percent of wild-type PP106-VM1-P *A. palmeri* plants survived the herbicide treatment, whilst survivorship was relatively high for heterozygous and homozygous plants at 73.2% and 81.9%, respectively. Whereas the difference in survivorship between heterozygous PS106-VM1-P and homozygous SS106-VM1-P mutant plants was not statistically significant, the differences were highly significant (P < 0.0001) when either heterozygous PS106-VM1-P or homozygous SS106-VM1-P mutant plants were compared with homozygous PP106-VM1-P plants (Table 4b). There was also clear evidence of a distinction in glyphosate survivorship between wild-type PP106-VM1-P and PP106-ApS *A. palmeri* plants when maintained at a high temperature (P = 0.002).

### 2.4. Prevalence of the P106S Mutation in 115 US A. palmeri Populations

Sequencing of a portion of the *EPSPS* gene from 115 Midwestern US populations identified several silent mutations, namely at codons 76, 98 and 106 (CCA or CCG), in addition to the ones previously found at positions 82 and 105 in VM1 and ApS *A. palmeri* samples. None of the 920 US *A. palmeri* individuals examined contained the glyphosate resistance-causing P106S *EPSPS* mutation.

## 3. Discussion

### 3.1. Mechanisms and Levels of Glyphosate Resistance Identified in VMI A. palmeri Population

Differential uptake was not a contributor to glyphosate resistance in the VM1 *A. palmeri* population, contrary to what was reported in a few *Lolium multiflorum*, *Digitaria insularis, Bidens pilosa*, *Amaranthus palmeri*, *Sorghum halepense, Chloris elata* and *Leptochloa virgata* populations [22,28,29,30,31,32]. Similarly, reduced movement to actively growing meristematic tissues identified in a wide range of *Conyza* spp., *Lolium* spp., *Amaranthus* spp., *Chloris elata* and *Eleusine indica* samples was not associated with resistance to glyphosate in VM1 [33,34,35,36]. The *EPSPS* copy number in *A. palmeri* VM1 was comparable to the standard sensitive population ApS, both clearly differentiated from the standard resistant population ApR characterised by the target gene duplication. This differs from most *A. palmeri* and *A. tuberculatus* populations investigated so far, whereby resistance to glyphosate could be explained by multiple *EPSPS* copies, initially thought to be dispersed throughout the genome and more recently, identified on extrachromosomal circular DNA as well [37,38,39,40,41,42]. There was a small but statistically significant 1.8-fold increase in *EPSPS* gene expression levels in *A. palmeri* VM1 compared with the standard sensitive population. Slightly higher levels of *EPSPS* expression were also found in some *Lolium rigidum* and *Conyza canadensis* populations, but were not sufficient to account for resistance to glyphosate [43,44]. Sequencing of the *EPSPS* gene detected a proline to serine amino acid change at codon 106 in the majority of VM1 individuals. The P106S mutation was previously observed alone in a number of *E. indica*, *Lolium* spp., *E. colona*, *C. canadensis*, *A. tuberculatus*, *A. palmeri*, and *Chloris virgata* populations and in combination with the T102I *EPSPS* mutation in *E. indica* and *B. pilosa* [22,32,45,46,47,48,49,50,51]. Proline 106 is not in direct interaction with the *EPSPS* substrate phosphoenol pyruvate (PEP) or the inhibitor glyphosate [52]. However, a substitution of proline 106 to a different amino acid results in structural changes in the active site affecting the binding of the PEP substrate and glyphosate [53]. The impact of the P106S mutation on glyphosate was clearly established using wild and mutant *EPSPS* extracted from whole plant tissues or expressed in *E. coli* [45,51]. At the whole plant level, however, the effect of amino acid changes at *EPSPS* codon 106 is rather unpredictable owing to the relatively low resistance intensity conferred by the mutations [54]. In the diploid *E. indica* species characterised by a single *EPSPS* copy, a 2- to 3-fold resistance increase due to the target site mutation was unambiguously demonstrated in two separate studies that employed wild and mutant P106S *EPSPS* plants from the same genetic background [55,56]. In both instances, resistance was partial even for homozygous mutant plants, with survivorship as low as 20% at the rate that killed the wild-type PP106 subpopulation. In contrast, mutations at *EPSPS* codon 106 were found not to endow sufficient levels of resistance to glyphosate in the polyploid *E. colona* species because of gene dilution effects [47]. Contrary to the *E. indica* and *E. colona* studies, the results were ambiguous for an *A. tuberculatus* population from Illinois, USA, as the P106S mutation was identified in both resistant and sensitive individuals [49]. In spite of the variable outcome associated with the P106S *EPSPS* mutation, most studies involving different weed species and populations settle on the importance of the proline to serine change (and other allelic variants) at *EPSPS* codon 106, citing the *E. coli* and *E. indica* researches [22,48,50,55,56,57]. Here, the impact of the P106S mutation was adequately determined using genetically comparable wild and mutant progenies originating from a cross between a large number of heterozygous PS106 plants. The resistance indices for heterozygous PS106 (RI = 4.3) and homozygous SS106 (RI = 7.1) plants were two to three times higher than those estimated for the equivalent *E. indica* mutant subpopulations [55,56]. However, resistance was still incomplete with some heterozygous and homozygous mutant individuals, respectively killed at one quarter and half the recommended field rate of glyphosate. It is noteworthy that the level of resistance conferred by homozygous SS106 mutant genotype was around half of that endowed by the 9-fold increase in *EPSPS* target copy and expression levels contained in the standard resistant population ApR. To our knowledge, this is the first side-by-side evaluation of the two different target-site-based mechanisms in conferring resistance to glyphosate. Comparison of wild-type PP106 genotypes from VM1-P and the standard sensitive population revealed additional underlying low levels of resistance (2.3-fold increase) that may be explained by the slightly higher levels of *EPSPS* expression in VM1. Concerted actions of different mechanisms, each contributing to relatively low levels of resistance, are expected and identified in several glyphosate-resistant populations [26,54].

### 3.2. Control of Wild and Mutant EPSPS Genotypes at Low and High Temperatures

Temperature is an important determinant for herbicide efficacy [58,59,60]. Whilst some studies have observed better control with glyphosate at high temperatures, others have reported the contrary, sometimes for the same species, depending on the populations and mechanisms of resistance involved [61,62]. For instance, better efficacy of glyphosate was noted on both sensitive and resistant *Ambrosia artemisiifolia* and *Ambrosia trifida* populations at 29 °C/17 °C day/night than at 20 °C/11 °C day/night, due to higher levels of herbicide movement to the meristem [63]. Pline et al. [64] also detected an increase of glyphosate translocation at high temperatures, resulting in a decrease of tolerance to the herbicide in genetically engineered soybean. In contrast, the efficacy of glyphosate decreased with increasing temperatures for some *L. rigidum*, *S. halepense K. scoparia, E. colona* and *C. canadensis* populations [62,65,66,67]. An extreme case is represented by a glyphosate-resistant *C. canadensis* biotype that was made sensitive when treated with glyphosate and kept at a low temperature [68]. Detailed NMR analysis of glyphosate in the vacuolar space detected higher levels of glyphosate sequestration at high temperatures accounting for resistance. When maintained at an unrealistic temperature of 11 °C for a relatively long period of time, the resistant *C. canadensis* population was completely killed due to lower ability of the individuals to sequester glyphosate in the vacuole [68]. Here, there was no difference in survivorship of the standard sensitive population kept at the low (20 °C/16 °C d/n) and the high (30 °C/26 °C d/n) temperatures, indicating that sufficient amounts of glyphosate were absorbed and translocated to meristematic tissues to kill the plants. A higher percentage of individuals survived in the second compared with the first experiment across all wild and mutant VMP-1 genotypes and temperature regimes, reflecting the test-to-test variation that can be observed in plant responses to herbicide applications [69,70]. Nevertheless, a similar trend of increased survivorship from wild PP106 to mutant SS106 through heterozygous PS106 plants was observed in the replicate test. When assessing genotypes between temperatures, a greater number of heterozygous and homozygous P106S mutant VM1-P individuals were controlled at low compared with the high temperature conditions, corroborating published results on *L. rigidum*, *K. scoparia*, *S. halepense, E. colona* and *C. canadensis* characterised by different mechanisms of glyphosate resistance [62,65,66,67]. At 30 °C/26 °C d/n, as many as 20% of wild-type PP106-VM1-P plants survived the glyphosate treatment, suggesting that under high-temperature conditions, weak resistance traits, represented here by slightly higher levels of *EPSPS* over-expression, are sufficient to allow the plants to escape the glyphosate treatment. Sub-optimally high temperature conditions also allowed the otherwise susceptible polyploid P106S mutant *E. colona* individuals to survive a commonly applied rate of glyphosate [47]. It is noteworthy that untreated plants across all genotypes maintained at the high temperature were on average three times taller than those kept at the low temperature. Given the impact of plant size on glyphosate efficacy, it is likely that faster growing and bigger plants at high temperatures, rather than reduced translocation of glyphosate, has allowed a larger proportion of individuals to overcome the herbicide application here and in other similar published studies [71,72,73].

### 3.3. Prevalence of the P106S Mutation in Native A. palmeri Populations 

*A. palmeri* in Argentina is suspected to have originated from the USA and introduced to the Latin American country via crop seed imports [19,26]. Yet, the P106S mutation identified in *A. palmeri* VM1 was not present in any of the 920 plants from 115 populations from Midwestern and Southern USA. The absence of the P106S mutation in the populations is in agreement with all the US samples investigated to date [37,40,41,74]. The presence and absence of the P106S change in VM1 and the US populations, respectively, may also be explained by the difference in cropping systems and management practices in the two American countries [75]. In Argentina, glyphosate-tolerant soybean is grown year after year in a quasi-monoculture. Glyphosate is frequently applied on relatively big *A. palmeri* plants due to large field sizes and rapid growth of the species [27]. Additionally, farmers and advisers often over-estimate the ability of herbicides to control large weeds. The over-reliance on glyphosate and application of the herbicide under sub-optimal plant size conditions favour the selection and accumulation of weak resistance mechanisms, such as the P106S mutation and low levels of *EPSPS* gene expression. In the USA, *A. palmeri* infesting corn, cotton and soybean crops are sprayed with glyphosate when they are relatively small for maximising weed control [13]. Furthermore, glyphosate is often complemented with pre-emergence products, and crop rotation is commonly practiced, allowing for a diverse set of herbicides for *A. palmeri* management [13,76]. The relatively more diverse chemical weed management practices in the USA are more likely to select against the weak P106S glyphosate-resistant trait. Contrasting glyphosate resistance mechanisms, very probably driven by different cropping and weed management strategies, are also reported for a few *A. palmeri* populations from Northern Mexico on the one hand and a sample from Mato Grosso, Brazil on the other hand [22,24]. In Northern Mexico, practical monocultures of glyphosate-tolerant cotton fields allowed for the selection of the P106S mutation (and low levels of reduced absorption and translocation) in three *A. palmeri* samples, similar to what is observed in the VM1 population [22]. In Mato Grosso, Brazil, where crop rotation is generally well-established, high levels of resistance to glyphosate in an *A. palmeri* population, accidentally introduced in the country by seed import from the USA, could be explained by a 50–179-fold increase in *EPSPS* gene copy numbers, similar to what is observed in all the native US *A. palmeri* samples [24]. 

### 3.4. Implications for A. palmeri Management in Argentina

Our study has revealed the presence of the relatively weak P106S target site resistance mechanism in VM1 which, under more adequate crop rotation and plant size conditions, does not appear to have been selected yet to cause glyphosate failures in native US *A. palmeri* samples [37,40,41,74]. Therefore, *A. palmeri* in Argentina should be targeted when they are small and vulnerable, using efficacious rates of glyphosate to avoid the selection and combination of weak glyphosate resistance mechanisms that will allow plants to survive and produce seeds. We also observed better levels of VM1 control at low compared with high temperature regimes, consistent with several other studies prompting their authors to recommend glyphosate application under cooler conditions [62,65,66,67,68]. Low temperatures of around 20 °C are sometimes encountered in the early morning in the soybean production areas in Argentina. However, the temperature is much higher for most of the day. In any case, a significant number of P106S mutant plants survived the glyphosate treatment at 20 °C d/16 °C n, implying that low temperatures alone will not be sufficient to overcome the weak resistance mechanisms in the VM1 population. For the effective management of the species, glyphosate should be complemented with other pre-emergence products, with overlapping residual activities and foliar herbicides belonging to other sites of actions [13,76]. Importantly, glyphosate-resistant soybean should be rotated with other summer crops, allowing use of alternative herbicides (e.g., atrazine and mesotrione in corn) for *A. palmeri* control [77]. Nonchemical *A. palmeri* control measures including tillage wherever possible, the use of cover crops, and hand and mechanical weeding should be encouraged [27]. Given the prevalent transmission of glyphosate resistance via contaminations, crop seeds that are weed-seed-free and clean farm machinery should be utilised [26,38]. A computer-based modelling approach integrating both chemical and nonchemical control methods could help design a long-term sustainable program for *A. palmeri* management in Argentina [78].

## 4. Materials and Methods 

### 4.1. Materials

Seeds from the suspected resistant *A. palmeri* population (VM1) were collected from a soybean field in Vicuña Mackenna, Cordoba, Argentina (Figure 5). The field was in continuous soybean production for five years and primarily managed with glyphosate at the time of seed sampling from *A. palmeri* survivors. A standard sensitive population (ApS) commercially sourced from Azlin Seed Service (USA) was used in all glasshouse and lab-based biological, physiological and molecular analysis (Figure 5). A known resistant population (ApR) originating from Georgia, USA and characterised by *EPSPS* gene-duplication and over-expression was included in all whole-plant dose–response and target gene-duplication and over-expression experiments (Figure 5). Additionally, 115 *A. palmeri* populations randomly sampled from southern and Midwestern US states were analysed for the main mechanism of glyphosate resistance identified in VM1. The 115 US *A. palmeri* populations were collected from field survivors during 2009–2016 from North Carolina (25), Kansas (18), Missouri (53), Alabama (10) and Arkansas (9) (Figure 5).

### 4.2. Plant Growth Conditions

Seeds from the different populations were separately sown into trays containing soil medium of a 1:1 ratio of compost and peat. The trays were maintained in controlled greenhouse conditions of 24/18 °C day/night, 65% relative humidity, and a photon flux density of approximately 250 µmol quanta m^−2^ s^−1^. When the emerged seedlings were 2 cm tall, they were individually transplanted into 75 mm diameter pots filled with the same aforementioned soil medium. The pots were irrigated and plants fertilised as necessary.

### 4.3. Initial Glyphosate Resistance Confirmation Test

ApS, ApR and VM1 *A. palmeri* plants, 8 cm tall and grown in the previously described conditions, were sprayed with a CO_2_-powered laboratory sprayer equipped with a flat-fan spray nozzle delivering a spray volume of 200 l ha^−1^. All three populations were treated with glyphosate (Roundup Weather MAX^®^) in a dose–response test. ApS plants were sprayed at 0, 25, 50, 100, 200, 400, 800 g ai ha^−1^, whilst ApR and VM1 individuals were treated at 0, 100, 200, 400, 800, 1600, 3200 g ai ha^−1^. Twenty replicate pots were tested for each population in a completely randomised design. Survivorship was recorded 21 days after treatment (DAT).

### 4.4. Mechanism of Resistance to Glyphosate

#### 4.4.1. *EPSPS* Gene Sequencing Around Known Glyphosate-Resistance-Causing Mutations

Twenty untreated plants each from ApS and VM1 populations were analysed for a potential gene mutation around *EPSPS* codons 102/106. A leaf segment of approximately 0.5 cm^2^ was sampled from each plant, placed in a Costar 96-well block (ThermoFisher Scientific, Leicestershire, UK) containing stainless steel beads (Qiagen, Manchester, UK), frozen to −80 °C and pulverised on a 2010 model Genogrinder (Spex Certiprep, Metuchen, NJ, USA). DNA was extracted from the ground material on a KingFisher^TM^ Flex Purification system (ThermoFisher Scientific, Leicestershire, UK) using the Wizard Magnetic 96 DNA Plant System kit (Promega, WI, USA). PCR targeted a 195 bp fragment using forward FW: 5’ATGTTGGACGCTCTCAGAACTCTTGGT3’ and reverse RV: 5’TGAATTTCCTCCAGCAACGGCAA3’ primers. The reaction was carried out in a 25 μL volume containing Ready-To-Go Taq beads (Amersham Biosciences, NJ, USA), 10–50 ng genomic DNA and primers at 20 pmol. PCR was conducted on a Master Cycle Gradient Thermocycler Model 96 (Eppendorf, UK) with the following conditions: a denaturation step at 95 °C for 5 min, followed by 30 cycles of 30 s at 95 °C, 30 s at 60 °C and 1 min at 72 °C. A final extension step for 10 min at 72 °C was also included. Direct Sanger sequencing (Genewiz LLC, USA) was carried out on 1 μL of neat Polymerase Chain Reaction product using the same forward PCR primer. Subsequently, the 40 (20 plants each from populations Aps and VM1) individual targeted *EPSPS* sequences were aligned and compared using Seqman software (DNASTAR Lasergene 10, DNASTAR, USA).

#### 4.4.2. Glyphosate Dose–Response Test on Precharacterised 106 *EPSPS* Genotypes

To determine the importance of the P106S target site mutation and other potential glyphosate resistance mechanisms in the VM1 population, a subsequent dose response study was carried out on precharacterised wild and mutant 106-*EPSPS* genotypes. For this purpose, 120 VM1 plants were grown in individual pots and sequenced around the 106 *EPSPS* codon, as described above. Fifty-two heterozygous VM1 individuals (26 males and 26 females) were selected and segregated into an isolated glasshouse bay with the same growth conditions (Figure 5). The plants were placed in a supporting cage, irrigated as required, allowed to cross freely among them and left to mature for four months. At maturity, flower heads from female plants were harvested and left to dry in a room at 12 °C and 12% relative humidity for 2 weeks. The flower heads were then threshed in an industrial seed processing machine (Model: Wintersteiger, Seed Processing, Holland) and cleaned to produce a fresh bag of seeds containing a mixture of genetically comparable wild-type homozygous PP106, heterozygous PS106 and homozygous SS106 mutant *EPSPS* plants (Figure 5). Individual 8 cm tall plants from the PS106 cross (denoted VM1-P) and the standard ApS and ApR populations were produced and utilised in a glyphosate dose–response test at 0, 12.5, 25, 50, 100, 200, 400, 800, 1600, 3200, 6400 g ai ha^−1^. Fourteen plants were tested per glyphosate rate for the standard sensitive ApS and resistant ApR populations. On the other hand, 96 randomly chosen plants were used per glyphosate rate for the heterogeneous population VM1-P (segregating into PP106, PS106 and SS106 plants) (Figure 5). Prior to herbicide application, a 0.5 cm^2^ leaf segment was sampled from all the 1056 (11 glyphosate rates x 96 plants) VM1-P individuals and genotyped into wild PP106, heterozygous PS106 and homozygous mutant SS106 subpopulations, as described in Section 4.4.1. The pots from the two standard sensitive and resistant populations and three characterised VM1-P subpopulations were arranged in a completely randomised design. Survivorship was recorded 21 days after treatment (DAT).

#### 4.4.3. *EPSPS* Target Gene-Duplication and Over-Expression

The VM1 population was assessed for potential *EPSPS* gene duplication and over-expression in comparison with the standard sensitive population ApS and resistant sample ApR (characterised by *EPSPS* gene duplication and over-expression) (Figure 5). Sixteen individually potted 8 cm plants were produced per population, as described in Section 4.2. A 0.5 cm^2^ leaf segment was harvested from each of the 48 untreated plants. These were placed in a Costar™ 96-well block (ThermoFisher Scientific, Leicestershire, UK) containing metal beads (ThermoFisher Scientific, Leicestershire, UK) and frozen at −80 °C. The frozen leaf materials of individual plants were ground on a 2010 model Genogrinder (Spex Certiprep, Metuchen NJ, USA), and lysis buffer (Promega, Madison, WI, USA) was added prior to DNA and RNA analysis. DNA was extracted from an aliquot of the ground material with the Wizard Magnetic DNA Plant System kit (Promega, Madison, WI, USA) and used in target gene-duplication studies. RNA was extracted using the RNeasy plant mini kit (Qiagen, Manchester, UK) and employed in target gene expression analysis. The RNA samples were cleaned from DNA using a DNAse treatment at 37 °C for 2 hours, followed by heat inactivation of the enzyme at 75 °C for 5 minutes. Corresponding cDNAs were generated from the RNA samples using High-Capacity cDNA Reverse Transcription Kit (ThermoFisher Scientific, Leicestershire, UK) according to the manufacturer’s recommendations.

Taqman assays were developed to determine the *EPSPS* gene copy number and expression level relative to the acetolactate synthase (*ALS*) and carbamoyl phosphate synthetase (*CPS*) genes. The primers were designed with Primer Express 3.0.1 (ThermoFisher Scientific, Leicestershire, UK) and based on GenBank entries KM438057, KT83338 and KM438047 for *ALS*, XM_010694055.2 for *CPS* and KC169784 and AY545657 for *EPSPS*. The primer names and sequences were as follows: *ALS*-forward 5’-TTCCTCGACATGAACAAGGTG-3’, *ALS*-reverse 5’-CCAACGCGTCCAGTAGCA-3’ and *ALS*-probe 5’-TTTTCGCTGCTGAAGGCTACGCTC-3’; *CPS*-forward 5’-TGCGGCAATTTTAAGAGCAT-3’, *CPS*-reverse 5’-GATGAGCTGAAGATTGAACAACCT-3’ and *CPS*-probe 5’- AGCTTCACTCCTAGCGATGCCTCCC-3’; *EPSPS*-forward 5’-GTCTAAAGCAACTTGGTTCAGATGT-3’, *EPSPS*-reverse 5’- CCCTGGAAGGCCTCCTTT-3’ and *EPSPS*-probe 5’- TGTTTTCTTGGCACAAATTGCCCTCC-3’.

The primers were diluted in 1xTE buffer, and corresponding efficiencies estimated using DNA extracted from the standard sensitive ApS plants. Real-time PCR reactions were set up in duplicate in a 10 μL volume reaction containing 1x Sigma JumpStart Taq ReadyMix, 300 nM of forward and reverse primers, 100 nM Probe and 3μL of either DNA or cDNA. QPCR was carried out on DNA and cDNA from the 16 replicate plants for each of the ApS, ApR and VM1 populations. All 288 reactions (3 populations x 16 individual plants x 3 genes x 2 technical replicates) were run on a single 384-well plate. Some additional wells were loaded with non-template controls and sensitive DNA and cDNA bulks, made up of a mixture of DNAs and cDNAs from the 16 ApS plants respectively. All samples were completely randomised and analysed on a QuantStudio 7 Flex Real-Time PCR System (ThermoFisher Scientific, Leicestershire, USA) with the following conditions: 95 °C for 5 min, followed by 40 cycles of 95 °C for 5 sec, followed by 60 °C for 30 s.

#### 4.4.4. Glyphosate Uptake and Translocation

VM1 and ApS plants grown at the 4-leaf stage, as described in Section 4.2 above, were treated with [phosphonomethylene] -^14^C glyphosate acid solution (0.45 MBq, with specific activity 4.729 MBq/mg) (Figure 5). Unlabelled glyphosate was supplemented to the radioactive solution to provide a treatment rate equivalent to 800 g ai ha^−1^ in a spray volume of 200 l ha^−1^. The glyphosate treatment was delivered in 20 x 0.2 μL microdroplets (4 μL total) across a 1 cm band in the middle of the adaxial surface of selected leaves to give 5000 Bq (total of 20 ug glyphosate, including 1.057 μg of ^14^C -glyphosate) per plant. The droplets were applied using a multidroplet applicator (Sartorius Biohit, Helsinski, Finland). Four replicate plants were treated for each population and time point. The plants were sampled at zero time (no more than 5 minutes after droplet application) for recovery comparisons and then at 24, 72 and 96 hours after treatment. Unabsorbed foliar surface residues were recovered by painting with cellulose acetate, and, once dry, the strips were removed and dissolved in 1 ml acetone. Radioactivity in the cellulose acetate strip was quantified by liquid scintillation counting (LSC) using a Perkin Elmer Tricarb 2900TR (Perkin Elmer, Waltham, MA, USA). The individual plants were freeze-dried and sectioned into treated area, meristem and rest of plant before sample oxidation using a Harvey OX 500 Biological Oxidiser with attached Zinsser robot (R. J. Harvey Instruments, Frankfurt, Germany). The individual plant sections were subsequently quantified by LSC. Percentage uptake was determined by the total amount of radioactivity detected in the plants x 100/total radioactivity applied (20 x microdroplets applied directly into glass scintillation vials). Relative herbicide translocation to the meristem and rest of plants was determined as: (sum of radioactivity from meristem or rest of plants) x 100/(total amount recovered from the meristem + rest of plant + treated area).

### 4.5. Influence of Temperature on the Efficacy of Glyphosate on Wild and Mutant EPSPS Genotypes

An experiment was conducted to determine whether glyphosate could be effective at controlling both wild and mutant plants at extreme low and high temperatures relevant to soybean production in Argentina. For this purpose, 8 cm tall APS and VM1-P (resulting from a cross between 52 PS106 VM plants) individually potted plants were produced as described in Section 4.2 (Figure 5). Forty ApS and 336 VM1-P plants were subsequently sprayed with a single recommended rate of glyphosate at 800 g ai ha^-1^, whilst 40 Aps and 60 VM1-P were left untreated for comparison. The plants were then divided into two equal lots (20 ApS and 30 VM1-P untreated individuals and 20 ApS and 168 VM1-P treated samples per lot) and kept in a growth cabinet at a low temperature of 20/16 °C day/night and a high temperature of 30/26 °C day/night, respectively. In both cases, the photon flux density was approximately 250 µmol quanta m^−2^ s^−1^ and there was 65% relative humidity. Prior to the glyphosate treatment, a 0.5 cm^2^ plant tissue from the 396 VM1-P progenies (336 treated and 60 untreated) was sampled and characterised at the 106 *EPSPS* codon position, as described in Section 4.4.1 above. The treated and untreated plants from the ApS and VM1-P populations were arranged in a completely randomised design within each temperature environment. Survivorship was recorded 21 days after treatment (DAT). The experiment was duplicated in time with slightly different plant numbers, as commanded by sample availability. The plants employed in the repeat experiment at each of the two temperatures were as follows: 20 ApS and 30 VM1-P individuals were untreated, and 28 ApS and 170 VM1-P samples were sprayed at 800 g ai ha^-1^ glyphosate. The detailed list of populations and genotypes tested across the two experiments is provided in Table 4a. 

### 4.6. Frequency of the P106S Mutation in a Large Number of A. palmeri from the USA

Seeds from 115 US *A. palmeri* populations were sown separately in trays, as described in Section 4.2 above. When the plants were at the 2-leaf stage, a 0.5 cm^2^ leaf section was sampled from eight individuals per population. All 920 plants (115 populations x 8 plants) were used for DNA, PCR and subsequently *EPSPS* gene sequencing analysis around the 106 codon, as fully described in Section 4.4.1.

### 4.7. Statistical Analysis

Plant survival data from the whole-plant dose–response test carried out on ApS, ApR and characterised PP106, PS106 and SS106 individuals (progenies from a cross between 52 PS106 individuals) were analysed by logit regression analysis (Finney, 1978), with identical slopes fitted to the regression lines for each population. LD_50_ estimates were obtained from the fitted regression lines, and resistance indices calculated as the ratio of the respective LD_50_ estimates. Both the LD_50_ and resistance index estimates are quoted with 95% confidence limits. A statistically significant (*P* = 0.05) difference between the populations may be concluded when the confidence interval for the resistance index does not include the value 1.

The C_T_ measurements for DNA and cDNA from the qPCR experiments were analysed separately. Prior to analysis, the C_T_ values for each plant and gene were averaged across the two technical replicates. These data were analysed by analysis of variance using the model:
(1)$$y_{\mathrm{ij}}=\mu +\gamma_{j}+\varepsilon_{\mathrm{ij}};$$
where $y_{\mathrm{ij}}$ denotes the difference between the average C_T_ value for the *EPSPS* gene and that of the *ALS* or *CPS* genes for plant i of population j, μ denotes the overall true mean, $\gamma_{j}$ denotes the effect of population j and $\varepsilon_{\mathrm{ij}}$ denotes the random error associated with plant i of population j. Comparisons between populations are then equivalent to carrying out a t-test using the pooled plant-to-plant variation within populations as the source of ‘error’ variation. The statistical significance of the population comparisons are summarised by a *p*-value, a value of 0.05 or less indicating a statistically significant result.

Glyphosate uptake and translocation measurements were analysed by factorial analysis of variance using the model:(2)$$y_{\mathrm{ijk}}=\mu +\gamma_{j}+\tau_{k}+{\left(\gamma \tau \right)}_{\mathrm{jk}}+\varepsilon_{\mathrm{ijk}};$$
where $y_{\mathrm{ijk}}$ denotes the response for replicate i of population j at time k, μ is the overall true mean response, $\gamma_{j}$ is the true effect of population j, $\tau_{k}$ is the true effect for time k, ${\left(\gamma \tau \right)}_{\mathrm{jk}}$ denotes the population-by-time interaction and $\varepsilon_{\mathrm{ijk}}$ is the random error associated with each individual response. Where there was evidence of a population-by-time interaction, populations were compared separately at each time point. Otherwise, populations were compared averaged across time.

For the survivorship of the glyphosate-treated wild and mutant plants at different temperatures, the data for each of the required population comparisons were structured as 2X2 contingency tables, the cells of which comprised the number of dead and surviving plants in each population. A test of the association between genotype and survival, taking into account the stratification of the data into two repeat tests is provided by the Cochran–Mantel–Haenszel (CMH) statistic, which is asymptotically distributed as a chi-square variable on 1 degree of freedom.

Logit analysis was carried out using Syngenta’s proprietary software. All other analyses were carried out using SAS version 9.4 (SAS Institute Inc., Cary, NC, USA).

## Figures and Tables

**Figure 1 plants-08-00512-f001:**
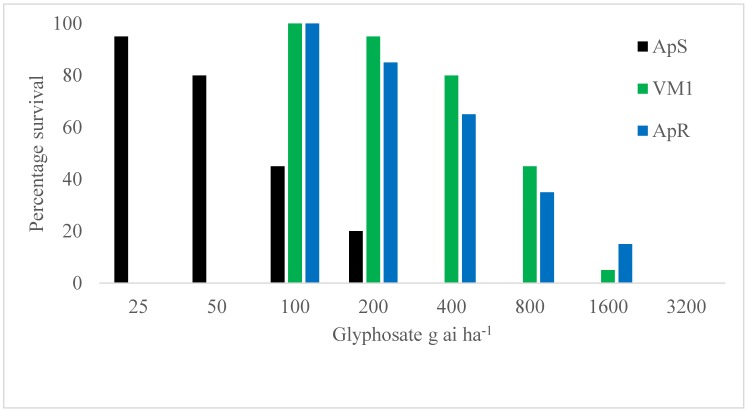
Glyphosate whole-plant dose–response test on VM1 compared with the known sensitive (ApS) and resistant (ApR: gene duplication) *A. palmeri* populations. Percentage survival assessed 21 days after glyphosate application.

**Figure 2 plants-08-00512-f002:**
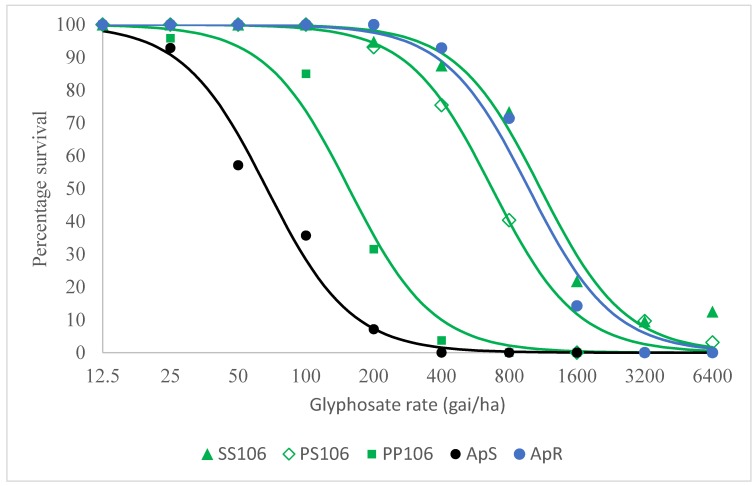
Glyphosate whole-plant dose–response assay on five *A. palmeri* populations: ApS (standard sensitive), ApR (known resistant due to *EPSPS* gene duplication) and three characterized subpopulations (homozygous wild-type PP106; heterozygous PS106 and homozygous mutant SS106) arising from a cross between 52 heterozygous PS106 individuals.

**Figure 3 plants-08-00512-f003:**
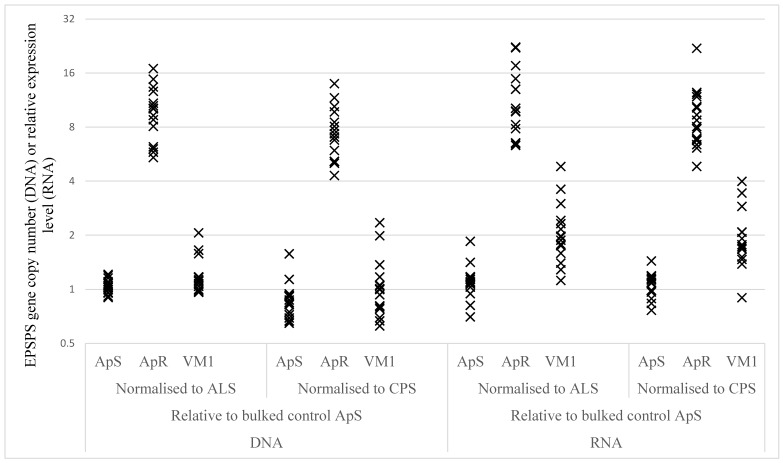
Scatter plot of *EPSPS* gene copy number and expression levels for each of the ApS, ApR and VM1 *A. palmeri* populations tested relative to bulked control samples. ALS: acetolactate synthase; CPS: carbamoyl phosphate synthetase.

**Figure 4 plants-08-00512-f004:**
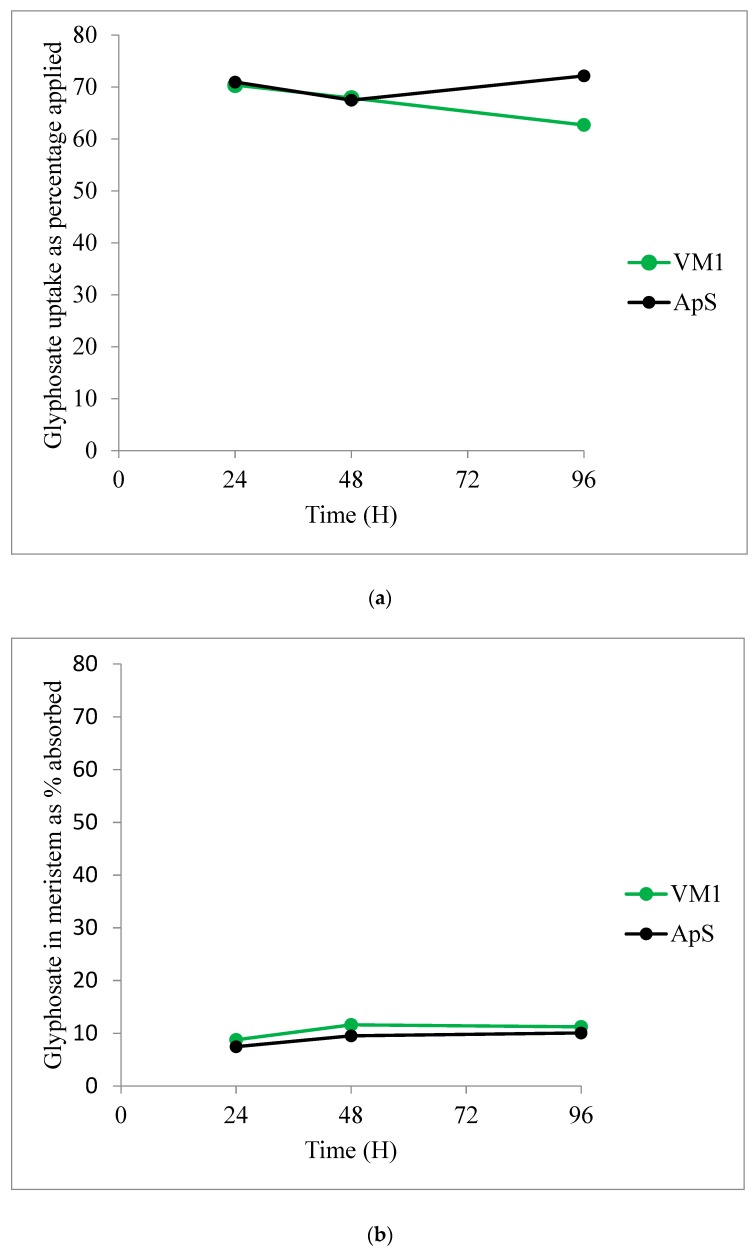
Relative uptake (**a**) of glyphosate and translocation to the meristem (**b**) in the standard sensitive ApS and VM1 *A. palmeri* populations.

**Figure 5 plants-08-00512-f005:**
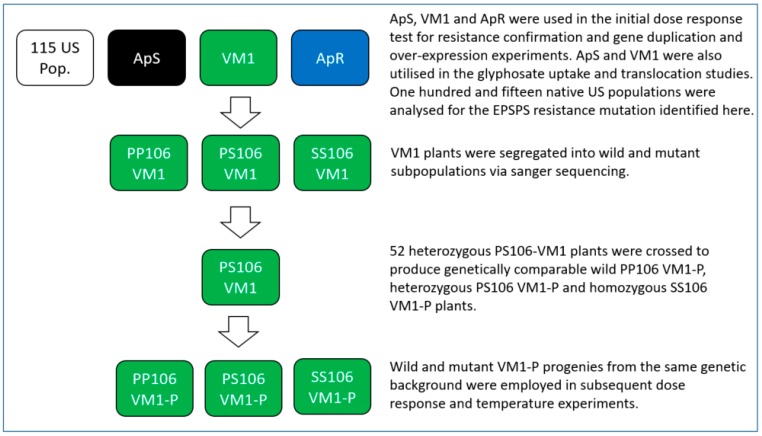
*Amaranthus palmeri* populations employed in this study.

**Table 1 plants-08-00512-t001:** LD_50_ values (**a**) and resistance indices (**b**) estimated from the glyphosate dose-response test on five plant groups: ApS (standard sensitive), ApR (known resistant due to *EPSPS* gene duplication) and three characterised subpopulations (homozygous wild-type PP106; heterozygous PS106 and homozygous mutant SS106), arising from a cross between 52 PS106 individuals (95% confidence limits in brackets).

**(a)**	
**Group**	**LD_50_**
SS106	1102 (857–1414)
PS106	672 (578–783)
PP106	155 (123–196)
ApS	67 (50–90)
ApR	977 (732–1305)
**(b)**	
**Comparison**	**Resistance Index**
SS106 vs PP106	7.1 (5.0–10.0)
PS106 vs PP106	4.3 (3.3–5.7)
PP106 vs ApS	2.3 (1.6–3.4)
SS106 vs ApS	16.4 (11.2–24.1)
PS106 vs ApS	10.0 (7.2–13.9)
ApR vs ApS	14.6 (9.7–22.0)

**Table 2 plants-08-00512-t002:** Average *EPSPS* gene copy number and expression relative to *ALS* and *CPS* genes for the ApS, ApR and VM1 populations.

Sample	Gene Comparison	ApS	ApR	VM1	ApR vs. ApS		VM1 vs. ApS	
					Ratio	P-Value	Ratio	P-Value
DNA	*EPSPS* vs *CPS*	0.85	7.35	0.99	8.68	<0.0001	1.17	0.1553
DNA	*EPSPS* vs *ALS*	1.04	9.43	1.18	9.07	<0.0001	1.13	0.1450
RNA	*EPSPS* vs *CPS*	1.05	8.81	1.87	8.37	<0.0001	1.78	<0.0001
RNA	*EPSPS* vs *ALS*	1.10	10.85	2.04	9.90	<0.0001	1.86	<0.0001

**Table 3 plants-08-00512-t003:** Means and standard errors for radiochemical recovered, expressed as percentage glyphosate absorbed in the standard sensitive ApS and VM1 *A. palmeri* populations.

Time after Treatment	24 h		48 h		72 h	
Population	ApS	VM1	ApS	VM1	ApS	VM1
Treated leaf	73.4 ± 1.8	68.2 ± 4.9	74.9 ± 7.4	62.7 ± 7.7	71.9 ± 10.3	39.7 ± 6.0
Meristem	7.5 ± 0.7	8.8 ± 1.2	9.5 ± 2.3	11.6 ± 2.7	10.1 ± 3.7	11.2 ± 1.1
Rest of plant	19.1 ± 2.0	23.0 ± 3.8	15.5 ± 5.3	25.7 ± 5.2	18.0 ± 6.6	49.1 ± 6.6

**Table 4 plants-08-00512-t004:** (**a**) Survivorship and corresponding genotypic data at *EPSPS* codon 106 for glyphosate-treated ApS and VM1-P plants maintained at low and high temperatures. (**b**) Comparison between plant genotypes based on *P*-values from the Cochran–Mantel–Haenszel test.

**(a)**					
**Growth Condition**	**Test**	**PP106-ApS**	**PP106-VM1-P**	**PS106-VM1-P**	**SS106-VM1-P**
20 °C day/16 °C night	1	0/20 = 0%	0/43 = 0%	15/91 = 16.5%	9/34 = 26.5%
	2	0/28 = 0%	4/31 = 12.9%	35/81 = 43.2%	34/58 = 58.6%
30 °C day/26 °C night	1	0/20 = 0%	4/35 = 11.4%	61/91 = 67.0%	32/43 = 74.4%
	2	0/28 = 0%	9/40 = 22.5%	73/92 = 79.3%	34/38 = 89.5%
**(b)**					
		**20 °C day/16 °C Night**		**30 °C day/26 °C Night**	
Comparison		CMH statistic	*P*-value	CMH statistic	*P*-value
PS106-VM1-P vs. PP106-VM1-P		16.7	<0.0001	68.9	<0.0001
SS106-VM1-P vs. PP106-VM1-P		29.1	<0.0001	65.2	<0.0001
SS106-VM1-P vs. PS106-VM1-P		4.8	0.0291	2.3	0.1264
PP106-VM1-P vs. PP106-ApS		3.8	0.0509	9.5	0.0020

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
