# Peer review of "Evolution of Target-Site Resistance to Glyphosate in an *Amaranthus palmeri* Population from Argentina and Its Expression at Different Plant Growth Temperatures"

_plants, 2019, doi:10.3390/plants8110512_

Round 1
Reviewer 1 Report
The modifications to be undertaken are marked in red colour and explained in the file attched bellow.

Author Response
Row 2 It may be suitable to change the title to have a better sense while reading it… e.g. Studies (Investigations) on the evolution and expression of glyphosate resistance mechanisms at different plant growth temperatures in an Amaranthus palmeri population from Argentina… or The evolution and expression….because like it is looks to me not clear! What on the evolution and expression…in Argentina? The name of the species should be written in the title complete as suggested not A. palmeri or use the common name and then the Latin one but all the time complete.
Response: A clearer title is provided as follows: Evolution of target-site resistance to glyphosate in an Amaranthus palmeri population from Argentina and its expression at different plant growth temperatures
Row 21 The name of the specie as it appears for the first time in the text should be written all e.g. Amaranthus palmeri then later could be written A. palmeri.
Response: Full name provided as requested
Row 28 The temperature should be written correctly 24 °C day/18 °C night.
Response: Changed as suggested
Row 30 Again the temperature should be written correctly at 30 °C day/26 °C night compared to 20 °C day/16 °C night.
Response: Changed as suggested
Row 104 Into figure 1 on the scale of the survival percentage should be written Percentage of survival.
Response: Percentage survival is the one which is grammatically correct
Row 105 At the text Figure 1. Should be written “Glyphosate whole plant dose response test on VM1 A. palmeri population compared to the known sensitive (ApS) and resistant (ApR) populations.
Response: A. Palmeri added to the title as suggested.
Row 106 Into the figure 1 text should be written The percentage of survivorship.
Response: Corrected as: Percentage survival for consistency with the Y axis of Figure 1
Row 130 Into figure 2 on the scale of the survival percentage should be written Percentage of survival.
Response: Percentage survival is grammatically correct
Row 131 Figure 2. Glyphosate whole plant dose response assay on A. palmeri populations ApS (standard sensitive), ApR (known 131 resistant due to EPSPS gene duplication) and three characterized sub-populations…
Response: Clarified as suggested
Row 161 Add to the phrase as suggested: ApR and the VM1 A. palmeri populations compared to the standard sensitive ApS sample.
Response: Clarified as suggested
Row 165 Add to the text as suggested: for each of the ApS, ApR and VM1 A. palmeri plants tested relative to bulked control samples.
Response: Added as requested
Row 167 Add to the text the species as suggested: ApR and VM1 A. palmeri plants.
Response: Added as requested
Row 188 meristematic tissues of the ApS and VM1 A. palmeri populations .
Response: Added as requested
Rows 211 and 232 Into figure 4 both parts 4a and 4b write in the same way the x lines Time(h) both bold or not and the same on the other part of the figure.
Response: Modified as requested
Row 236 Write better the title of the table e.g. : Means and standard errors for radiochemicals recovered expressed as a percentage of glyphosate absorbed in the standard sensitive ApS and VM1 A. palmeri populations. Use the same type of writing for the time maybe bold all as the left part of the table is also bold.
Response: Rewritten as requested
Row 239 All untreated plants survived at a low temperature of 20 °C/16 °C day/night
Response: Modified as requested
Row 240 of 30 °C/26 °C day/night conditions
Response: Modified as requested
Row 242 All standard sensitive ApS A. palmeri plants were killed when treated with glyphosate
Response: Added as requested
Row 249 All ApS A. palmeri plants treated with the single…
Response: Added as requested
Row 251 type PP106-VM1-P A. palmeri plants survived
Response: Modified as requested
Row 259 glyphosate treated ApS and VM1-P A. palmeri plants maintained…
Into the table 4a add the symbol of degree “ °“ before Celsius where needed.
Response: Added as requested
Note: Into the text at part 2.3. Relative control of wild and mutant plants at low and high temperatures
First describe 4a results then should follow 4b not first 4b and then 4a. Or re-write the text to be clear.
Response: Note that 4a is first described at line 190 and 4b at line 193 in the revised manuscript
Row 261 Table 4b?. Comparison between A. palmeri plant genotypes based on…
Response: 4b added as requested
Row 266 in VM1 and ApS A. palmeri samples.
Response: Added as requested
Row 269 3.1. Mechanisms and levels of glyphosate resistance identified in VMI A. palmeri population.
Response: Added as requested
Row 270 Differential uptake was not a contributor to glyphosate resistance in VM1 A. palmeri population contrary…
Response: Modified as requested
Row 274 delete extra space between Amaranthus spp., and Chloris elata…
Response: Extra space deleted
Row 275 The EPSPS copy number in VM1 population of A. palmeri was comparable
Response: Added as requested
Row 281 gene expression levels in VM1 A. palmeri compared to the standard sensitive population.
Response: Added as requested
Row 294 Delete extra the space before …In the diploid ….
Response: Extra space deleted
Row 303 Delete extra the space before the phrase In spite of the variable outcome…
Response: Extra space deleted
Row 312 Delete extra the space before the phrase: It is noteworthy….
Response: Extra space deleted
Row 318 Delete the space before the phrase: Concerted actions of different…
Response: Extra space deleted
Row 323 populations at 29 °C/17 °C day/night than at 20 °C/11 °C
Response: Added as requested
Row 335 temperature of 11°C for a relatively long
Response: Added as requested
Row 338 at the low (20 °C/16 °C d/n) and the high (30 °C/26 °C d/n) temperatures indicating….
Response: Added as requested
Row 348 At 30 °C/26 °C d/n, as many as 20% of wild type PP106-VM1-P plants survived…
Response: Added as requested
Row 361 Yet, the P106S mutation identified in VM1 A. palmeri population was not…
Response: A. palmeri added as requested
Row 366 Delete the extra space before the phrase: In Argentina, glyphosate-tolerant soybean…
Response: Extra space deleted
Row 380 similar to what is observed in VM1 population [22].
Response: Added as requested
Row 394 Low temperatures of around 20 °C are sometimes encountered..
Response: Symbol added as requested
Row 397 20 °C d/16 °C n, implying that low temperatures alone will…
Response: Added as requested
Row 398 resistance mechanisms in VM1 A. palmeri population.
Response: Added as requested
Row 425 24/18 °C day/night, 65% relative humidity…
Response: Symbol added as requested
Rows 424 and 427 Delete the extra space before the phrases. keep the same uniformity all over the text into the paper.
Response: Extra space deleted
Row 430 Could be better to add the species name after VM1 plants e.g. VM1 A. palmeri plants.
Response: Added as requested
Because you repeat the same word aforementioned some rows before can be changed to a synonymous e.g. previously described conditions…
Response: Modified as requested
Row 439 Twenty untreated plants each from ApS and VM1 populations were analysed…
Response: Populations added as requested
Row 444 delete Ltd after UK
Response: Ltd deleted as requested
Row 445 Write the name of the primer before the sequence using forward …..name 5’ATGTTGGACGCTCTCAGAACTCTTGGT3’ and name reverse….
Response: Primers named as requested
Row 452 product using the same forward primer as for PCR.
Response: Modified as requested
Row 457 resistance mechanisms in VM1 population, a subsequent…
Response: Modified as requested
Row 479 VM1 population…
Response: Added as requested
Row 483 These were placed in a Costar™ 96-Well block (maybe ThermoFisher Scientific, USA… all the times you use it in the text or the manufacturer exact and correct name) containing
Response: ThermoFisher Scientific employed for consistency across the paper
Row 484 beads (Fisher Scientific, USA…) and frozen at -80 °C
Response: Symbol added as requested
Row 485 and lysis buffer (Promega, WI, USA) was…
Response: WI, USA added as requested
Row 488 RNA was extracted using the RNeasy plant mini kit (Qiagen, Hilden, Germany)…
Response: Hilden, Germany added as requested
Row 490 at 37 °C for 2 hours followed by heat inactivation of the enzyme at 75 °C for 5 minutes…
Response: Symbol added as requested
Row 492 Kit (ThermoFisher Scientific, USA) according to the manufacturer’s recommendations.
Response: Modified as requested
Row 495 …with Primer Express 3.0.1 software (Thermofisher Scientific, USA) and based on GenBank entries…
Response: Added as requested
Row 497 The primers names and sequences are as follows….
Response: Added as requested
Row 504 The primers were diluted in 1xTE buffer and their corresponding efficiencies estimated….
Response: Buffer added as requested
Row 506 in a 10 μl reaction volume containing 1x Sigma JumpStart Taq ReadyMix….
Response: Volume added
Row 513 (ThermoFisher Scientific, USA?) under the following conditions: 95 °C for 5 min followed by 40 cycles of 95 °C for 5 sec
Response: Symbol added as requested
Row 514 followed by 60 °C for 30 sec.
Response: Symbol added as requested
Row 516 VM1 and ApS plants grown at the 4-leaf stage as described in section 2.2 above were treated with…
Response: Modified as requested
Row 522 The droplets were applied using a Sartorius Biohit multidroplet applicator (manufacturer, country).
Response: Address details added
Row 528 a Perkin Elmer Tricarb 2900TR (manufacturer, country).
Response: Address details added
Row 530 with attached Zinnser robot (R. J. Harvey Instruments, country).
Response: Address details added
Row 544 at a low temperature of 20/16 °C day/night and a high…
Response: Symbol added as requested
Row 547 a 0.5 cm2 plant tissue from the 396 VM1-P progenies (336 treated and 60 untreated) was taken and…
Response: Sampled instead of taken is an appropriate verb for use here
Row 545 temperature of 30/26°C day/night respectively.
Response: Symbol added as requested
Row 558 I think that at the phrase: When the plants reached the 2-leaf stage, a 0.5 cm2 leaf section was sampled from 8 individuals per population change leaf section was sampled to a portion of leaf of about 0.5 cm2 was taken from 8 individuals per population for DNA extraction.
Response: Sentence modified for clarity
Rows 559 to 561 Re-write the phrase: All 920 plants (115 populations x 8 plants) were extracted for DNA and subsequent PCR and EPSPS gene sequencing analysis conducted around the 106 codon as described in section 2.4.1. to be more clear. For example: All 920 plants (115 populations x 8 plants) were used for DNA, PCR and subsequently EPSPS gene sequencing analysis conducted around the 106 codon as fully described in section 2.4.1.
Response: Sentence modified as requested
Row 594 Logit analysis was carried out using Syngenta’s proprietary software (name of the software, country).
Response: The proprietary software does not have a name
Row 595 carried out using SAS version 9.4. (SAS Institute Inc., USA).
Response: Modified as requested
Reviewer 2 Report
The topic of this manuscript is very interesting. Authors, examined the resistance of an Amaranthus palmeri population to glyphosate in Argentina. I consider that the manuscript contains information's that deserve to be published after major revision.
I provide below a few suggestions that, if the authors decide to implement into the paper, the paper will improved.
Comments
Title: The title of article should be revised since the topic of this article was not only the expression of glyphosate resistance at different plant growth temperatures.
Abstract: The temperatures units should be corrected.
Introduction: The introduction section is well written.
Material and methods: This section is well written.
Minor corrections:
The temperature units should be corrected. Line 431: Authors should corrected the term CO2 as CO2 Lines 434-435: The authors should revised this phrase (Twenty replicate pots were tested….). Line 434: Pots size should be mentioned.
Results
Units should be added in some figures and tables. Section 2.1: This section should be revised. Poor presentation of the results. The results that presented in Figure 1 should be presented in a figure with 2 diagrams: an diagram about ApS population and one diagram about the resistance populations since applied different doses per population. Section 2.1.1. The differences in the nucleotide sequences for VM1 populations and ApS populations should be presented in figure. Table 1 should be improved Section 2.2.4: Poor presentation of the results. The authors should show the statistical significant differences between the two populations in figure 4 and Table 3. Lines 192: H should be corrected as h. Lines 256-262: The numbering of tables should be corrected (Table 4 and Table 4a)
Discussion:
This section should be slightly revised. Lines 367-370: Authors should explain why the application of glyphosate on large A. palmeri populations favour the selection and accumulation of week resistance mechanisms. Also, authors should explain why in Argentina the farmers applied the herbicides in large A. palmeri plants. Lines 374-376: The authors should revised this phrase. Line 402: The authors should mention which herbicides can be applied for control of this weed in other summer crops.
Author Response
The topic of this manuscript is very interesting. Authors, examined the resistance of an Amaranthus palmeri population to glyphosate in Argentina. I consider that the manuscript contains information's that deserve to be published after major revision.
I provide below a few suggestions that, if the authors decide to implement into the paper, the paper will improved.
Comments
Title: The title of article should be revised since the topic of this article was not only the expression of glyphosate resistance at different plant growth temperatures.
Response: Title modified as requested
Abstract: The temperatures units should be corrected.
Response: Temperature unit added
Introduction: The introduction section is well written.
Response: Thank you for your appreciation
Material and methods: This section is well written.
Minor corrections:
The temperature units should be corrected.
Response: Temperature symbols added
Line 431: Authors should corrected the term CO2 as CO2
Response: Modified as requested
Lines 434-435: The authors should revised this phrase (Twenty replicate pots were tested….). Line 434: Pots size should be mentioned.
Response: Pot size mentioned at line 430
Results
Units should be added in some figures and tables.
Response: Units added as requested
Section 2.1: This section should be revised. Poor presentation of the results. The results that presented in Figure 1 should be presented in a figure with 2 diagrams: an diagram about ApS population and one diagram about the resistance populations since applied different doses per population.
Response: The dose response use split rates for the standard sensitive and resistant populations. This is a customary practice to ensure a dose response is obtained for both sensitive and resistant populations. As the plants were tested in the same experiment it is better to show the results on one figure only for ease of comparison.
Section 2.1.1. The differences in the nucleotide sequences for VM1 populations and ApS populations should be presented in figure.
Response: Only a small EPSPS fragment around the binding site of glyphosate was analysed and the important nucleotide and codon changes are provided in the revised manuscript (CCA to TCA). An additional figure will not add much to the result.
Table 1 should be improved
Response: Table 1 has been reformatted accordingly
Section 2.2.4: Poor presentation of the results. The authors should show the statistical significant differences between the two populations in figure 4 and Table 3.
Response: The important measurement here is the amount of glyphosate that reached the meristem between the standard sensitive and resistant population VM1. As mentioned in the text: Similar amounts (P = 0.41) of radiochemicals were recovered in the meristematic tissues of the ApS and VM1 A. palmeri populations. Many other pairwise combinations among different plant sections are possible and as mentioned in the text, the differences are not statistically significant. An additional table showing negative results will not add value to the manuscript.
Lines 192: H should be corrected as h.
Response: Corrected as suggested
Lines 256-262: The numbering of tables should be corrected (Table 4 and Table 4a)
Response: Corrected as suggested
Discussion:
This section should be slightly revised. Lines 367-370: Authors should explain why the application of glyphosate on large A. palmeri populations favour the selection and accumulation of weak resistance mechanisms.
Response: As explained in the subsequent sentence, treating large instead of small plants amounts to application of glyphosate under sub-optimal conditions.
Also, authors should explain why in Argentina the farmers applied the herbicides in large A. palmeri plants.
Response: This is explained in the revised version of the manuscript in the following manner: Glyphosate is frequently applied on relatively big A. palmeri plants due to large field sizes and rapid growth of the species. Additionally, farmers and advisers often over-estimate the ability of herbicides to control large weeds.
Lines 374-376: The authors should revised this phrase.
Response: We believe that the sentence is well-constructed and clear: The relatively more diverse chemical weed management practices in the USA are more likely to select against the weak P106S glyphosate resistant trait.
Line 402: The authors should mention which herbicides can be applied for control of this weed in other summer crops.
Response: The alternative herbicides are mentioned in the revised manuscript.
Reviewer 3 Report
line 44 - stay consistent with either the common or Latin name, the authors go back and forth
line 48 - what do you mean by seedling growth? continual germination?
Materials and methods should be moved to after introduction.
line 413 - while this line is stated to be susceptible, it is quite often not. Was another susceptible line used?
line 435 - only 20 plants total per population? what about temporal replications? where the additional 115 populations collected tested as well in the dose response?
line 436 - no biomass or other measurements? one measurement of survivorship is not enough
line 100 - first mention of field rates, stay consistent with g ai ha-1
Figures should not use red and green bars, this makes it difficult for people who are color blind. Figures need SE bars and significance test results reported on them.
At what level of survivorship is resistance confirmed? is 65% enough?
Not fully convinced by your results. Also, there is some disconnect between the methods and results.
Author Response
line 44 - stay consistent with either the common or Latin name, the authors go back and forth
Response: This is correct and of common practice
line 48 - what do you mean by seedling growth? continual germination?
Response: Seedling growth is the rate that the seedling grow. Continuous germination is something else
Materials and methods should be moved to after introduction.
Response: The recommended style for Plants is that the introduction is followed by results
line 413 - while this line is stated to be susceptible, it is quite often not. Was another susceptible line used?
Response: Only the ApS line was employed as sensitive population for comparison. The sensitivity of this population was confirmed in the initial glyphosate dose response test (Figure 1) whereby the population was killed at half the recommended field rate (400 g ai/ha) of glyphosate. The same observation was made for the dose response test with characterised genotypes (Figure 2).
line 435 - only 20 plants total per population? what about temporal replications?
Response: The test was repeated in time with characterised genotypes (see section 4.4.2)
Were the additional 115 populations collected tested as well in the dose response?
Response: The additional 115 populations were only analysed for the EPSPS codon at position 106.
line 436 - no biomass or other measurements? one measurement of survivorship is not enough
Response: Collecting biomass for over 1400 individual plants is tedious and space time consuming and do not add more regarding the set objectives and conclusion reached in this study.
line 100 - first mention of field rates, stay consistent with g ai ha-1
Response: g ai ha-1 was consistently used throughout the manuscript
Figures should not use red and green bars, this makes it difficult for people who are color blind. Figures need SE bars and significance test results reported on them.
Response: Colours changed on Figures as requested. Error bars do not add value to the figure as the statistical significance of the results is already provided as confidence intervals in the corresponding tables.
At what level of survivorship is resistance confirmed? is 65% enough?
Response: Resistance is a serious issue and 65% survival will likely cause significant yield loss given that a single plant per 30 cm of row reduced grain yield by as much as 64%, according to a field study in Arkansas (line 48-49).
Not fully convinced by your results. Also, there is some disconnect between the methods and results.
Response: As pointed out by the three other reviewers, The data generated as part of this study allowed us to reach clear conclusions on the 3 set objectives (i) mechanism of resistance to glyphosate in the A. palmeri population VM1 (ii) relative expression of the mechanisms of resistance identified in VM1 using pre-characterised wild and mutant progenies applied with glyphosate and allowed to grow at different temperatures relevant to soybean cropping systems in Argentina (iii) assessment of the prevalence of the target-site resistance mutation identified in VM1 in a large number of native US A. palmeri.
Reviewer 4 Report
Overall this was an excellent paper outlining a series of experiments to provide information on a case of glyphosate resistance from Argentina. It has enough unique features to make it a very useful addition to the literature on the topic of glyphosate resistance.
My only comments are very minor ones really. I spent quite some time reading the paper thoroughly and I found myself confused initially about all of the different populations being studied. Although the captions of some of the tables and graphs helped explain the difference between the various groups of individuals, the authors may wish to look at adding in a table or paragraph somewhere clearly explaining the difference between the various populations used in this work, or the series of abbreviations put together to describe the various types of plants. I think I have it clear in my head now but it shouldn’t need so much puzzling over the finer details to get a clear picture.
I am also a little unsure whether there are errors in the section numbering mentioned in the methods section. For example, on Line 475, there is a reference to section 2.4.1, yet no such section exists. Likewise, there is mention of section 2.2 on Line 516 and again on Line 540, which might have been correct, but I wasn’t sure they were.
Likewise for Table 4, there is Table 4a above a Table 4, with the second table presumably meant to be Table 4b.
I couldn’t see that Figure 4 was needed, as the same information for both graphs is shown better in Table 3, as the standard errors get included, as do the differences between the populations at 72 h, whereas the two graphs miss these differences.
Given the results in Table 4a and comments made at the end of the discussion, and as figures should stand alone from the text, it would be good to mention in the caption for Figure 1 the size or age of the plants when sprayed for this experiment and the average temperature they were kept under.
The standard of English and formatting was generally very good. However, the formatting did vary at times, probably due to the different authors involved, and one example of this was how the temperature was listed using different formats, eg compare Lines 335 and 338 with Lines 449-450, then another method in Lines 544-545. Botanical names were used for weed species almost everywhere in the paper, then local common names crept in at times, such as Line 326. Occasionally the terminology used was not good (possibly typos), with examples being:
Line 73: “import from” should be “imported from”
Line 182: “both of the two latter ratios were” should be “the latter two ratios were both”
Line 464: “Then after, the flowers heads were” should be “The flower heads were then”
Line 471: “On the other hand,’ would probably be improved by “However,”
Line 551: “as commanded by sample availability” would probably be better as “depending on sample availability”
Author Response
Overall this was an excellent paper outlining a series of experiments to provide information on a case of glyphosate resistance from Argentina. It has enough unique features to make it a very useful addition to the literature on the topic of glyphosate resistance.
My only comments are very minor ones really. I spent quite some time reading the paper thoroughly and I found myself confused initially about all of the different populations being studied. Although the captions of some of the tables and graphs helped explain the difference between the various groups of individuals, the authors may wish to look at adding in a table or paragraph somewhere clearly explaining the difference between the various populations used in this work, or the series of abbreviations put together to describe the various types of plants. I think I have it clear in my head now but it shouldn’t need so much puzzling over the finer details to get a clear picture.
Response: As suggested, an additional Figure 5 is provided in the manuscript to clearly indicate populations and subpopulations employed in the different studies.
I am also a little unsure whether there are errors in the section numbering mentioned in the methods section. For example, on Line 475, there is a reference to section 2.4.1, yet no such section exists. Likewise, there is mention of section 2.2 on Line 516 and again on Line 540, which might have been correct, but I wasn’t sure they were.
Response: The numbering has been corrected accordingly
Likewise for Table 4, there is Table 4a above a Table 4, with the second table presumably meant to be Table 4b.
Response: Corrected accordingly
I couldn’t see that Figure 4 was needed, as the same information for both graphs is shown better in Table 3, as the standard errors get included, as do the differences between the populations at 72 h, whereas the two graphs miss these differences.
Response: We suggest maintain Figure 4 as it is more telling than a table.
Given the results in Table 4a and comments made at the end of the discussion, and as figures should stand alone from the text, it would be good to mention in the caption for Figure 1 the size or age of the plants when sprayed for this experiment and the average temperature they were kept under.
Response: We believe the figures are good as such and that any addition will make them too busy and unclear.
The standard of English and formatting was generally very good. However, the formatting did vary at times, probably due to the different authors involved, and one example of this was how the temperature was listed using different formats, eg compare Lines 335 and 338 with Lines 449-450, then another method in Lines 544-545.
Response: The symbol for degree centigrade was added throughout the manuscript as suggested
Botanical names were used for weed species almost everywhere in the paper, then local common names crept in at times, such as Line 326.
Response: Scientific names were used in the revised manuscript
Occasionally the terminology used was not good (possibly typos), with examples being:
Line 73: “import from” should be “imported from”
Response: Seed import is the correct grammatical form to be used here
Line 182: “both of the two latter ratios were” should be “the latter two ratios were both”
Response: Modified accordingly
Line 464: “Then after, the flowers heads were” should be “The flower heads were then”
Response: Modified accordingly
Line 471: “On the other hand,’ would probably be improved by “However,”
Response: On the other hand is right here
Line 551: “as commanded by sample availability” would probably be better as “depending on sample availability”
Response: As commanded by sample availability is correct here.
Round 2
Reviewer 2 Report
The authors following the comments during the reviewing process improved the manuscript. Thus, this article can be accepted for publication on this journal.
Reviewer 3 Report
Improved although I do not agree with some of your responses. Much clarification that you provided in the reviewer comments should have been addressed in the manuscript itself.